# PC-Fairness: A Unified Framework for Measuring Causality-based Fairness

**Yongkai Wu**
University of Arkansas
yw009@uark.edu

**Lu Zhang**
University of Arkansas
lz006@uark.edu

**Xintao Wu**
University of Arkansas
xintaowu@uark.edu

**Hanghang Tong**
University of Illinois at Urbana-Champaign
htong@illinois.edu

## Abstract

A recent trend of fair machine learning is to define fairness as causality-based notions which concern the causal connection between protected attributes and decisions. However, one common challenge of all causality-based fairness notions is identifiability, i.e., whether they can be uniquely measured from observational data, which is a critical barrier to applying these notions to real-world situations. In this paper, we develop a framework for measuring different causality-based fairness. We propose a unified definition that covers most of previous causality-based fairness notions, namely the path-specific counterfactual fairness (PC fairness). Based on that, we propose a general method in the form of a constrained optimization problem for bounding the path-specific counterfactual fairness under all unidentifiable situations. Experiments on synthetic and real-world datasets show the correctness and effectiveness of our method.

## 1 Introduction

Fair machine learning is now an important research field which studies how to develop predictive machine learning models such that decisions made with their assistance fairly treat all groups of people irrespective of their protected attributes such as gender, race, etc. A recent trend in this field is to define fairness as causality-based notions which concern the causal connection between protected attributes and decisions. Based on Pearl's structural causal models [8], a number of causality-based fairness notions have been proposed for capturing fairness in different situations, including total effect [19, 16, 20], direct/indirect discrimination [19, 16, 7, 20], and counterfactual fairness [5, 14, 15, 9].

One common challenge of all causality-based fairness notions is identifiability, i.e., whether they can be uniquely measured from observational data. As causality-based fairness notions are defined based on different types of causal effects, such as total effect on interventions, direct/indirect discrimination on path-specific effects, and counterfactual fairness on counterfactual effects, their identifiability depends on the identifiability of these causal effects. Unfortunately, in many situations these causal effects are in general unidentifiable, referred to as unidentifiable situations [12]. Identifiability is a critical barrier for the causality-based fairness to be applied to real applications. In previous works, simplifying assumptions are proposed to evade this problem [5, 19, 4]. However, these simplifications may severely damage the performance of predictive models. In [20] the authors propose a method to bound indirect discrimination as the path-specific effect in unidentifiable situations, and in [14] a method is proposed to bound counterfactual fairness. Nevertheless, the tightness of these methods is not analyzed. In addition, it is not clear whether these methods can be applied to other unidentifiable situations, and more importantly, a combination of multiple unidentifiable situations.

In this paper, we propose a framework for handling different causality-based fairness notions. We first propose a general representation of all types of causal effects, i.e., the path-specific counterfactual effect, based on which we define a unified fairness notion that covers most previous causality-based fairness notions, namely the path-specific counterfactual fairness (PC fairness). We summarize all unidentifiable situations that are discovered in the causal inference literature. Then, we develop a constrained optimization problem for bounding the PC fairness, which is motivated by the method proposed in [2] for bounding confounded causal effects. The key idea is to parameterize the causal model using so-called response-function variables, whose distribution captures all randomness encoded in the causal model, so that we can explicitly traverse all possible causal models to find the tightest possible bounds. In the experiments, we evaluate the proposed method and compare it with previous bounding methods using both synthetic and real-world datasets. The results show that our method is capable of bounding causal effects under any unidentifiable situation or combinations. When only path-specific effect or counterfactual effect is considered, our method provides tighter bounds than methods in [20] or [14]. The proposed framework settles a general theoretical foundation for causality-based fairness. We make no assumption about the hidden confounders so that hidden confounders are allowed to exist in the causal model. We also make no assumption about the data generating process and whether the observation data is generated by linear or non-linear functions would not introduce bias into our results. We only assume that the causal graph is given, which is a common assumption in structural causal models.

**Relationship to other work.** In [3], the author introduces the term "path-specific counterfactual fairness", which states that a decision is fair toward an individual if it coincides with the one that would have been taken in a counterfactual world in which the sensitive attribute along the unfair pathways were different. They develop a correction method called PSCF for eliminating the individual-level unfair information contained in the observations while retaining fair information. Compared to [3], we formally define a general fairness notion which, besides the individual-level fairness, is also applied to fairness in any sub-group of the population. In addition, we further consider the identifiability issue in causal inference that is inevitably brought by conditioning on the individual level. Unidentifiable situation means that there exist two causal models which exactly agree with the same observational distribution (hence cannot be distinguished using statistic methods such as maximum likelihood), but lead to very different causal effects. In our paper, we address various unidentifiable situations by developing a general bounding method. The authors in [6] study the conditional path-specific effect and develop a complete identification algorithm with the application to the problem of algorithmic fairness. Similar to our proposed notion, their notion is also quantified via conditional distributions over the interventional variant. However, the conditional path-specific effect generalizes the conditional causal effect, where the factual condition is assumed to be "non-contradictory" (such as age in measuring the effect of smoking on lung cancer) [12]. The path-specific counterfactual effect, on the other hand, generalizes the counterfactual effect, where the factual condition can be contradictory to the observation. Formally, in the conditional path-specific effect, the condition is performed on the pre-intervention distribution, but in the path-specific counterfactual effect, the condition is performed on the post-intervention distribution.

## 2 Preliminaries

In our notations, an uppercase denotes a variable, e.g., $X$; a bold uppercase denotes a set of variables, e.g., $\mathbf{X}$; and a lowercase denotes a value or a set of values of the variables, e.g., $x$ and $\mathbf{x}$.

### 2.1 Causal Model and Causal Graph

**Definition 1** (Structural Causal Model [8]). *A structural causal model $\mathcal{M}$ is represented by a quadriple $\langle \mathbf{U}, \mathbf{V}, \mathbf{F}, P(\mathbf{U}) \rangle$ where*

1. *$\mathbf{U}$ is a set of exogenous variables that are determined by factors outside the model.*
2. *$P(\mathbf{U})$ is a joint probability distribution defined over $\mathbf{U}$.*
3. *$\mathbf{V}$ is a set of endogenous variables that are determined by variables in $\mathbf{U} \cup \mathbf{V}$.*
4. *$\mathbf{F}$ is a set of structural equations from $\mathbf{U} \cup \mathbf{V}$ to $\mathbf{V}$. Specifically, for each $V \in \mathbf{V}$, there is a function $f_V \in \mathbf{F}$ mapping from $\mathbf{U} \cup (\mathbf{V} \backslash V)$ to $V$, i.e., $v = f_V(\mathsf{pa}_V, u_V)$, where $\mathsf{pa}_V$ is a realization of a set of endogenous variables $\mathsf{PA}_V \in \mathbf{V} \setminus V$ that directly determines $V$, and $u_V$ is a realization of a set of exogenous variables that directly determines $V$.*

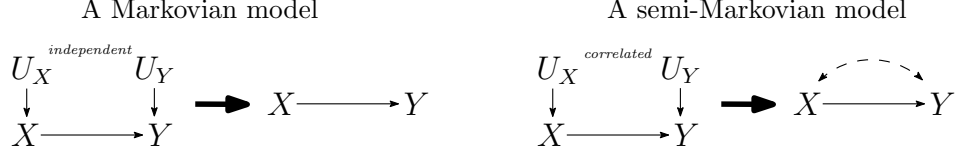

Figure 1: Causal graphs of a Markovian model and a semi-Markovian models

In general, $f_V(\cdot)$ can be an equation of any type. In some cases, people may assume that $f_V(\cdot)$ is of a specific type, e.g., the nonlinear additive function if $v = f_V(\mathsf{pa}_V) + \mathbf{u}_V$. On the other hand, if all exogenous variables in $\mathbf{U}$ are assumed to be mutually independent, then the causal model is called a *Markovian model*; otherwise, it is called a *semi-Markovian model*. In this paper, we don't make assumptions about the type of equations and independence relationships among exogenous variables.

The causal model $\mathcal{M}$ is associated with a causal graph $\mathcal{G} = \langle \mathcal{V}, \mathcal{E} \rangle$ where $\mathcal{V}$ is a set of nodes and $\mathcal{E}$ is a set of edges. Each node of $\mathcal{V}$ corresponds to a variable of $\mathbf{V}$ in $\mathcal{M}$. Each edge in $\mathcal{E}$, denoted by a directed arrow $\rightarrow$, points from a node $X \in \mathbf{U} \cup \mathbf{V}$ to a different node $Y \in \mathbf{V}$ if $f_Y$ uses values of $X$ as input. A *causal path* from $X$ to $Y$ is a directed path which traces arrows directed from $X$ to $Y$. The causal graph is usually simplified by removing all exogenous variables from the graph. In a Markovian model, exogenous variables can be directly removed without loss of information. In a semi-Markovian model, after removing exogenous variables we also need to add dashed bi-directed edges between the children of correlated exogenous variables to indicate the existence of unobserved common cause factors, i.e., hidden confounders. Examples are demonstrated in Figure 1.

## 2.2 Causal Effects

Quantitatively measuring causal effects in the causal model is facilitated with the *do*-operator [8] which forces some variable $X$ to take certain value $x$, formally denoted by $do(X = x)$ or $do(x)$. In a causal model $\mathcal{M}$, the intervention $do(x)$ is defined as the substitution of structural equation $X = f_X(\mathsf{PA}_X, U_X)$ with $X = x$. For an observed variable $Y$ ($Y \neq X$) which is affected by the intervention, its interventional variant is denoted by $Y_x$. The distribution of $Y_x$, also referred to as the post-intervention distribution of $Y$ under $do(x)$, is denoted by $P(Y_x = y)$ or simply $P(y_x)$.

By using the *do*-operator, the total causal effect is defined as follows.

**Definition 2** (Total Causal Effect [8]). *The total causal effect of the value change of $X$ from $x_0$ to $x_1$ on $Y = y$ is given by*

$$\mathrm{TCE}(x_1, x_0) = P(y_{x_1}) - P(y_{x_0}).$$

The total causal effect is defined as the effect of $X$ on $Y$ where the intervention is transferred along all causal paths from $X$ to $Y$. If we force the intervention to be transferred only along a subset of all causal paths from $X$ to $Y$, the causal effect is then called the path-specific effect, defined as follows.

**Definition 3** (Path-specific Effect [1]). *Given a causal path set $\pi$, the $\pi$-specific effect of the value change of $X$ from $x_0$ to $x_1$ on $Y = y$ through $\pi$ (with reference $x_0$) is given by*

$$\mathrm{PE}_\pi(x_1, x_0) = P(y_{x_1|\pi, x_0|\bar{\pi}}) - P(y_{x_0}),$$

*where $P(Y_{x_1|\pi, x_0|\bar{\pi}})$ represents the post-intervention distribution of $Y$ where the effect of intervention $do(x_1)$ is transmitted only along $\pi$ while the effect of reference intervention $do(x_0)$ is transmitted along the other paths.*

Definition 2 and 3 consider the average causal effect over the entire population without any prior observations. If we have certain observations about a subset of attributes $\mathbf{O} = \mathbf{o}$ and use them as conditions when inferring the causal effect, then the causal inference problem becomes a *counterfactual inference* problem meaning that the causal inference is performed on the sub-population specified by $\mathbf{O} = \mathbf{o}$ only. Symbolically, the distribution of $Y_x$ conditioning on factual observation $\mathbf{O} = \mathbf{o}$ is denoted by $P(y_x|\mathbf{o})$. The counterfactual effect is defined as follows.

**Definition 4** (Counterfactual Effect [12]). *Given a factual condition $\mathbf{O} = \mathbf{o}$, the counterfactual effect of the value change of $X$ from $x_0$ to $x_1$ on $Y = y$ is given by*

$$\mathrm{CE}(x_1, x_0|\mathbf{o}) = P(y_{x_1}|\mathbf{o}) - P(y_{x_0}|\mathbf{o}).$$

Table 1: Connection between previous fairness notions and PC fairness

| Description | References | Relating to PC fairness |
|---|---|---|
| Total effect | [19, 16] | $\mathbf{O} = \emptyset$ and $\pi = \Pi$ |
| (System) Direct discrimination | [19, 7, 16] | $\mathbf{O} = \emptyset$ or $\{S\}$ and $\pi = \pi_d = \{S \to \hat{Y}\}$ |
| (System) Indirect discrimination | [19, 7, 16] | $\mathbf{O} = \emptyset$ or $\{S\}$ and $\pi = \pi_i \subset \Pi$ |
| Individual direct discrimination | [17] | $\mathbf{O} = \{S, \mathbf{X}\}$ and $\pi = \pi_d = \{S \to \hat{Y}\}$ |
| Group direct discrimination | [18] | $\mathbf{O} = \mathbf{Q} = \mathsf{PA}_Y \backslash \{S\}$ and $\pi = \pi_d = \{S \to \hat{Y}\}$ |
| Counterfactual fairness | [5, 9, 14] | $\mathbf{O} = \{S, \mathbf{X}\}$ and $\pi = \Pi$ |
| Counterfactual error rate | [15] | $\mathbf{O} = \{S, Y\}$ and $\pi = \pi_d$ or $\pi_i$ |

## 3 Path-specific Counterfactual Fairness

In this section, we define a unified fairness notion for representing different causality-based fairness notions. The key component of our notion is a general representation of causal effects. Consider an intervention on $X$ which is transmitted along a subset of causal paths $\pi$ to $Y$, conditioning on observation $\mathbf{O} = \mathbf{o}$. Based on that, we define path-specific counterfactual effect as follows.

**Definition 5** (Path-specific Counterfactual Effect)**.** *Given a factual condition $\mathbf{O} = \mathbf{o}$ and a causal path set $\pi$, the path-specific counterfactual effect of the value change of $X$ from $x_0$ to $x_1$ on $Y = y$ through $\pi$ (with reference $x_0$) is given by*

$$\mathrm{PCE}_\pi(x_1, x_0 | \mathbf{o}) = P(y_{x_1 | \pi, x_0 | \bar{\pi}} | \mathbf{o}) - P(y_{x_0} | \mathbf{o}).$$

In the context of fair machine learning, we use $S \in \{s^+, s^-\}$ to denote the protected attribute, $Y \in \{y^+, y^+\}$ to denote the decision, and $\mathbf{X}$ to denote a set of non-protected attributes. The underlying mechanism of the population over the space $S \times \mathbf{X} \times Y$ is represented by a causal model $\mathcal{M}$, which is associated with a causal graph $\mathcal{G}$. A historical dataset $\mathcal{D}$ is drawn from the population, which is used to construct a predictor $h : \mathbf{X}, S \to \hat{Y}$. The causal model for the population over space $S \times \mathbf{X} \times \hat{Y}$ can be considered the same as $\mathcal{M}$ except that function $f_Y$ is replaced with a predictor $h$. We use $\Pi$ to denote all causal paths from $S$ to $\hat{Y}$ in the causal graph.

Then, we define the path-specific counterfactual fairness based on Definition 5.

**Definition 6** (Path-specific Counterfactual Fairness (PC Fairness))**.** *Given a factual condition $\mathbf{O} = \mathbf{o}$ where $\mathbf{O} \subseteq \{S, \mathbf{X}, Y\}$ and a causal path set $\pi$, predictor $\hat{Y}$ achieves the PC fairness if $\mathrm{PCE}_\pi(s_1, s_0 | \mathbf{o}) = 0$ where $s_1, s_0 \in \{s^+, s^-\}$. We also say that $\hat{Y}$ achieves the $\tau$-PC fairness if $\left| \mathrm{PCE}_\pi(s_1, s_0 | \mathbf{o}) \right| \leq \tau$.*

We show that previous causality-based fairness notions can be expressed as special cases of the PC fairness. Their connections are summarised in Table 1, where $\pi_d$ contains the direct edge from $S$ to $\hat{Y}$, and $\pi_i$ is a path set that contains all causal paths passing through any redlining attributes (i.e., a set of attributes in $\mathbf{X}$ that cannot be legally justified if used in decision-making). Based on whether $\mathbf{O}$ equals $\emptyset$ or not, the previous notions can be categorized into the ones that deal with the system level ($\mathbf{O} = \emptyset$) and the ones that have certain conditions ($\mathbf{O} \neq \emptyset$). Based on whether $\pi$ equals $\Pi$ or not, the previous notions can be categorized into the ones that deal with the total causal effect ($\pi = \Pi$), the ones that consider the direct discrimination ($\pi = \pi_d$), and the ones that consider the indirect discrimination ($\pi = \pi_i$).

In addition to unifying the existing notions, the notion of PC fairness also resolves new types of fairness that the previous notions cannot do. One example is individual indirect discrimination, which means discrimination along the indirect paths for a particular individual. Individual indirect discrimination has not been studied yet in the literature, probably due to the difficulty in definition and identification. However, it can be directly defined and analyzed using PC fairness by letting $\mathbf{O} = \{S, \mathbf{X}\}$ and $\pi = \pi_i$.

## 4 Measuring Path-specific Counterfactual Fairness

In this section, we develop a general method for bounding the path-specific counterfactual effect in any unidentifiable situation. In the causal inference field, researchers have studied the reasons

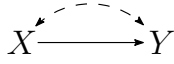

Figure 2: The "bow graph".

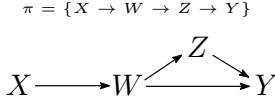

Figure 3: The "kite graph".

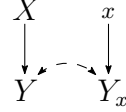

Figure 4: The "w graph".

for unidentifiability under different cases. When $\mathbf{O} = \emptyset$ and $\pi \subset \Pi$, the reason for unidentifiability can be the existence of the "kite graph" (see Figure 3) in the causal graph [1]. When $\mathbf{O} \neq \emptyset$ and $\pi = \Pi$, the reason for unidentifiability can be the existence of the "w graph" (see Figure 4) [11]. In any situation, as long as there exists a "hedge graph" (where the simplest case is the "bow graph" as shown in Figure 2), then the causal effect is unidentifiable [12]. Obviously, all above unidentifiable situations can exist in the path-specific counterfactual effect.

Our method is motivated by [2] which formulates the bounding problem as a constrained optimization problem. The general idea is to parameterize the causal model and use the observational distribution $P(\mathbf{V})$ to impose constraints on the parameters. Then, the path-specific counterfactual effect of interest is formulated as an objective function of maximization or minimization for estimating its upper or lower bound. The bounds are guaranteed to be tight as we traverse all possible causal models when solving the optimization problem. Thus, a byproduct of the method is a unique estimation of the path-specific counterfactual effect in the identifiable situation.

For presenting our method, we first introduce a key concept called the response-function variable.

## 4.1 Response-function Variable

Response-function variables are proposed in [2] for parameterizing the causal model. Consider an arbitrary endogenous variable denoted by $V \in \mathbf{V}$, its endogenous parents denoted by $\mathsf{PA}_V$, its exogenous parents denoted by $U_V$, and its associated structural function in the causal model denoted by $v = f_V(\mathsf{pa}_V, u_V)$. In general, $U_V$ can be a variable of any type with any domain size, and $f_V$ can be any function, making the causal model very difficult to be handled. However, we can note that, for each particular value $u_V$ of $U_V$, the functional mapping from $\mathsf{PA}_V$ to $V$ is a particular deterministic response function. Thus, we can map each value of $U_V$ to a deterministic response function. Although the domain size of $U_V$ is unknown which might be very large or even infinite, the number of different deterministic response functions is known and limited, given the domain sizes of $\mathsf{PA}_V$ and $V$. This means that the domain of $U_V$ can be divided into several equivalent regions, each corresponding to the same response function. As a result, we can transform the original non-parameterized structural function to a limited number of parameterized functions.

Formally, we represent equivalent regions of each endogenous variable $V$ by the *response-function variable* $R_V = \{0, \cdots, N_V - 1\}$ where $N_V = |V|^{|\mathsf{PA}_V|}$ is the total number of different deterministic response functions mapping from $\mathsf{PA}_V$ to $V$ ($N_V = |V|$ if $V$ has no parent). Each value $r_V$ represents a pre-defined response function. We also denote the mapping from $U_V$ to $R_V$ as $r_V = \ell_V(u_V)$. Then, for any $f_V(\mathsf{pa}_V, u_V)$, it can be re-formulated as

$$f_V(\mathsf{pa}_V, u_V) = f_V(\mathsf{pa}_V, \ell_V^{-1}(r_V)) = f_V \circ \ell_V^{-1}(\mathsf{pa}_V, r_V) = g_V(\mathsf{pa}_V, r_V),$$

where $g_V$ is the composition of $f_V$ and $\ell_V^{-1}$, and denotes the response functions represented by $r_V$. We denote the set of all response-function variables by $\mathbf{R} = \{R_V : V \in \mathbf{V}\}$.

Next, we show how joint distribution $P(\mathbf{v})$ can be expressed as a linear function of $P(\mathbf{r})$. According to [13], $P(\mathbf{v})$ can be expressed as the summation over the probabilities of certain values $\mathbf{u}$ of $\mathbf{U}$ that satisfy following corresponding requirements: for each $V \in \mathbf{V}$, we must have $f_V(\mathsf{pa}_V, u_V) = v$ where $v, \mathsf{pa}_V$ are specified by $\mathbf{v}$ and $u_V$ is specified by $\mathbf{u}$. In other words, denoting by $V(\mathbf{u})$ the value that $V$ would obtain if $\mathbf{U} = \mathbf{u}$, we have $P(\mathbf{v}) = \sum_{\mathbf{u}:\mathbf{V}(\mathbf{u})=\mathbf{v}} P(\mathbf{u})$. Then, by mapping from $\mathbf{U}$ to $\mathbf{R}$, we accordingly obtain $P(\mathbf{v}) = \sum_{\mathbf{r}:\mathbf{V}(\mathbf{r})=\mathbf{v}} P(\mathbf{r})$, where for each $V \in \mathbf{V}$, $V(\mathbf{r}) = v$ means that $g_V(\mathsf{pa}_V, r_V) = v$. As a result, by defining an indicator function

$$\mathbb{I}(v; \mathsf{pa}_V, r_V) = \begin{cases} 1 & \text{if } g_V(\mathsf{pa}_V, r_V) = v, \\ 0 & \text{otherwise}, \end{cases}$$

we obtain

$$P(\mathbf{v}) = \sum_{\mathbf{r}} P(\mathbf{r}) \prod_{V \in \mathbf{V}} \mathbb{I}(v; \mathsf{pa}_V, r_V), \qquad (1)$$

which is a linear expression of $P(\mathbf{r})$.

**Example 1.** Consider the causal graph shown in Figure 1 with two endogenous variables $X$ and $Y$, and two exogenous variables $U_X$ and $U_Y$ with unknown domains. Assume that both $X$ and $Y$ are binary, i.e., $X \in \{x_0, x_1\}$ and $Y \in \{y_0, y_1\}$, and denote their response variables as $R_X$ and $R_Y$. For $Y$, since there are a total number of $2^2 = 4$ response functions, response-function variable $R_Y$ and response function $g_Y$ can be defined as follows:

$$r_Y = \ell_Y(u_Y) = \begin{cases} 0 \text{ if } f_Y(x_0, u_Y) = y_0, f_Y(x_1, u_Y) = y_0; \\ 1 \text{ if } f_Y(x_0, u_Y) = y_0, f_Y(x_1, u_Y) = y_1; \\ 2 \text{ if } f_Y(x_0, u_Y) = y_1, f_Y(x_1, u_Y) = y_0; \\ 3 \text{ if } f_Y(x_0, u_Y) = y_1, f_Y(x_1, u_Y) = y_1. \end{cases} \qquad g_Y(x, r_Y) = \begin{cases} y_0 \text{ if } r_Y = 0; \\ y_0 \text{ if } x = x_0, r_Y = 1; \\ y_1 \text{ if } x = x_1, r_Y = 1; \\ y_1 \text{ if } x = x_0, r_Y = 2; \\ y_0 \text{ if } x = x_1, r_Y = 2; \\ y_1 \text{ if } r_Y = 3. \end{cases}$$

Similarly, response-function variable $R_X$ and response function $g_X$ can be defined as

$$r_X = \ell_X(u_X) = \begin{cases} 0 & \text{if } f_X(u_X) = x_0; \\ 1 & \text{if } f_X(u_X) = x_1. \end{cases} \qquad\qquad g_X(r_X) = \begin{cases} x_0 & \text{if } r_X = 0; \\ x_1 & \text{if } r_X = 1. \end{cases}$$

As a result, the joint distribution over $X, Y$ is given by

$$P(x, y) = \sum_{r_X, r_Y} P(r_X, r_Y) \mathbb{I}(x; r_X) \mathbb{I}(y; x, r_Y).$$

## 4.2 Expressing Path-specific Counterfactual Fairness

For bounding the path-specific counterfactual effect, i.e., $\text{PCE}_\pi(s_1, s_0|\mathbf{o}) = P(\hat{y}_{s_1|\pi, s_0|\bar{\pi}}|\mathbf{o}) - P(\hat{y}_{s_0}|\mathbf{o})$, we also apply response-function variables to express it. We focus on the expression of $P(\hat{y}_{s_1|\pi, s_0|\bar{\pi}}|\mathbf{o})$, and the expression of $P(\hat{y}_{s_0}|\mathbf{o})$ can be similarly obtained as a simpler case. Similar to the previous section, we first express $P(\hat{y}_{s_1|\pi, s_0|\bar{\pi}}|\mathbf{o})$ as the summation over the probabilities of certain values of $\mathbf{U}$ that satisfy corresponding requirements. However, as described below, the requirements are much more complicated than previous ones due to the integration of intervention, path-specific effect, and counterfactual.

Firstly, since the path-specific counterfactual effect is under a factual condition $\mathbf{O} = \mathbf{o}$, values $\mathbf{u}$ must satisfy that $\mathbf{O}(\mathbf{u}) = \mathbf{o}$, i.e., for each $O \in \mathbf{O}$, we must have $f_O(\mathsf{pa}_O, u_O) = o$. Secondly, the path-specific counterfactual effect is transmitted only along some path set $\pi$. According to [20], for the variables of $\mathbf{X}$ that lie on both $\pi$ and $\bar{\pi}$, referred to as *witness variables/nodes* [1], we need to consider two sets of values, one obtained by treating them on $\pi$ and the other obtained by treating them on $\bar{\pi}$. Formally, non-protected attributes $\mathbf{X}$ are divided into three disjoint sets. We denote by $\mathbf{W}$ the set of witness variables, denote by $\mathbf{A}$ the set of non-witness variables on $\pi$, and denote by $\mathbf{B}$ the set of non-witness variables on $\bar{\pi}$. A simple example is given in Figure 5. We denote the interventional variant of $\mathbf{A}$ by $\mathbf{A}_{s_1|\pi}$, the interventional variant of $\mathbf{B}$ by $\mathbf{B}_{s_0|\bar{\pi}}$, the interventional variant of $\mathbf{W}$ treated on $\pi$ by $\mathbf{W}_{s_1|\pi}$, and the interventional variant of $\mathbf{W}$ treated on $\bar{\pi}$ by $\mathbf{W}_{s_0|\bar{\pi}}$. Then, $P(\hat{y}_{s_1|\pi, s_0|\bar{\pi}}|\mathbf{o})$ can be written as

$$P(\hat{y}_{s_1|\pi, s_0|\bar{\pi}}|\mathbf{o}) = \sum_{\mathbf{a}, \mathbf{b}, \mathbf{w}_1, \mathbf{w}_0} P(\hat{Y}_{s_1|\pi, s_0|\bar{\pi}} = y, \mathbf{A}_{s_1|\pi} = \mathbf{a}, \mathbf{B}_{s_0|\bar{\pi}} = \mathbf{b}, \mathbf{W}_{s_1|\pi} = \mathbf{w}_1, \mathbf{W}_{s_0|\bar{\pi}} = \mathbf{w}_0 \mid \mathbf{o}).$$

To obtain the above joint distribution, in addition to $\mathbf{O}(\mathbf{u}) = \mathbf{o}$, values $\mathbf{u}$ must also satisfy that:

1. $\mathbf{A}_{s_1|\pi}(\mathbf{u}) = \mathbf{a}$, which means for each $A \in \mathbf{A}$, we must have $f_A(\mathsf{pa}_A^1, u_A) = a$, where $\mathsf{pa}_A^1$ means that if $\text{PA}_A$ contains $S$ or any witness node $W$, its value is specified by $s_1$ or $w_1$ if edge $S/W \rightarrow Y$ belongs to a path in $\pi$, and specified by $s_0$ or $w_0$ otherwise;
2. $\mathbf{B}_{s_0|\bar{\pi}}(\mathbf{u}) = \mathbf{b}$, which means for each $B \in \mathbf{B}$, we must have $f_B(\mathsf{pa}_B^0, u_B) = b$, where $\mathsf{pa}_B^0$ means that if $\text{PA}_B$ contains $S$ or any witness node $W$, its value is specified by $s_0$ or $w_0$;

3. $\mathbf{W}_{s_1|\pi}(\mathbf{u}) = \mathbf{w}_1$, which means for each $W \in \mathbf{W}$, we must have $f_W(\mathsf{pa}_W^1, u_W) = w_1$;
4. $\mathbf{W}_{s_0|\pi}(\mathbf{u}) = \mathbf{w}_0$, which means for each $W \in \mathbf{W}$, we must have $f_W(\mathsf{pa}_W^0, u_W) = w_0$.

Then, by mapping from $\mathbf{U}$ to $\mathbf{R}$, we can obtain the requirements for $\mathbf{R}$ accordingly. Finally, denoting the values of $\mathbf{R}$ that satisfy $\mathbf{O}(\mathbf{r}) = \mathbf{o}$ by $\mathbf{r_o}$, we obtain

$$P(\hat{y}_{s_1|\pi,s_0|\bar{\pi}}|\mathbf{o}) =$$

$$\sum_{\substack{\mathbf{a},\mathbf{b},\mathbf{w}_1 \\ \mathbf{w}_0,\mathbf{r}\in\mathbf{r_o}}} \frac{P(\mathbf{r})}{P(\mathbf{o})}\mathbb{I}(\hat{y};\mathsf{pa}_{\hat{Y}}^1,r_{\hat{Y}})\prod_{A\in\mathbf{A}}\mathbb{I}(a;\mathsf{pa}_A^1,r_A)\prod_{B\in\mathbf{B}}\mathbb{I}(b;\mathsf{pa}_B^0,r_B)\prod_{W\in\mathbf{W}}\mathbb{I}(w_1;\mathsf{pa}_W^1,r_W)\mathbb{I}(w_0;\mathsf{pa}_W^0,r_W), \quad (2)$$

which is still a linear expression of $P(\mathbf{r})$.

Similarly, we can obtain

$$P(\hat{y}_{s_0}|\mathbf{o}) = \sum_{\mathbf{v}',\mathbf{r}\in\mathbf{r_o}} \frac{P(\mathbf{r})}{P(\mathbf{o})}\mathbb{I}(\hat{y};\mathsf{pa}_{\hat{Y}},r_{\hat{Y}}) \prod_{V\in\mathbf{V}'} \mathbb{I}(v;\mathsf{pa}_V,r_V), \quad (3)$$

where $\mathbf{V}' = \mathbf{V}\backslash\{S,Y\}$.

**Example 2.** Consider causal graphs shown in Figures 2, 3, 4 and following unidentifiable causal effects: total causal effect $\text{TCE}(x_1,x_0)$ in Figure 2, path-specific effect $\text{PE}_\pi(x_1,x_0)$ in Figure 3, and counterfactual effect $\text{CE}(x_1,x_0|x_0,y_0)$ in Figure 4. By similarly defining response functions as in Example 1, for Figure 2 with $\mathbf{R} = \{R_X, R_Y\}$, we have

$$\text{TCE}(x_1,x_0) = \sum_{r_X,r_Y} P(r_X,r_Y)\mathbb{I}(y;x_1,r_Y) - \sum_{r_X,r_Y} P(r_X,r_Y)\mathbb{I}(y;x_0,r_Y),$$

for Figure 3 with $\mathbf{R} = \{R_X, R_W, R_Z, R_Y\}$, we have

$$\text{PE}_\pi(x_1,x_0) = \sum_{z,w_1,w_0,\mathbf{r}} P(\mathbf{r})\mathbb{I}(y;z,w_0,r_Y)\mathbb{I}(z;w_1,r_Z)\mathbb{I}(w_1;x_1,r_W)\mathbb{I}(w_0;x_0,r_W)$$

$$- \sum_{z,w,\mathbf{r}} P(\mathbf{r})\mathbb{I}(y;z,w,r_Y)\mathbb{I}(z;w,r_Z)\mathbb{I}(w;x_0,r_W),$$

for Figure 4 with $\mathbf{R} = \{R_X, R_Y\}$, we have

$$\text{CE}(x_1,x_0) = \sum_{r_X,r_Y\in\mathbf{r_o}} \frac{P(r_X,r_Y)}{P(x_0,y_0)}\mathbb{I}(y;x_1,r_Y) - \sum_{r_X,r_Y\in\mathbf{r_o}} \frac{P(r_X,r_Y)}{P(x_0,y_0)}\mathbb{I}(y;x_0,r_Y).$$

Note that in Figures 2, the total causal effect is identifiable if $U_X$ and $U_Y$ are independent. This is reflected in our formulation such that when $R_X$ and $R_Y$ are independent, we have $P(y_{x_1}) = \sum_{r_X,r_Y} P(r_X)P(r_Y)\mathbb{I}(y;x_1,r_Y) = P(y|x_1)$, which can be directly measured from observational data. Similar phenomenons can be observed in other identifiable situations.

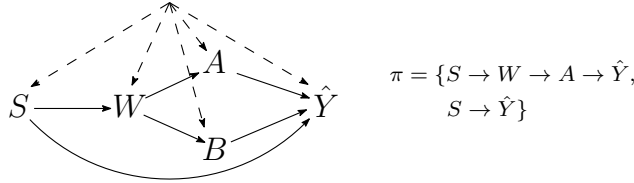

$$\pi = \{S \to W \to A \to \hat{Y},$$
$$S \to \hat{Y}\}$$

Figure 5: A causal graph with unidentifiable path-specific counterfactual fairness.

**Example 3.** Consider a causal graph shown in Figure 5, and the path-specific counterfactual effect $\text{PCE}_\pi(s_1,s_0|\mathbf{o})$ where $\pi = \{S \to \hat{Y}, S \to W \to A \to \hat{Y}\}$ and $\mathbf{o} = \{s_0, w', a', b'\}$. Any pair of exogenous variables can be correlated. Response-function variables are given by $\mathbf{R} = \{R_S, R_W, R_A, R_B, R_{\hat{Y}}\}$. By similarly defining response functions as in Example 1, we can obtain

$$P(\hat{y}_{s_1|\pi,s_0|\bar{\pi}}|\mathbf{o}) = \sum_{\substack{a,b,w_1,w_0 \\ \mathbf{r}\in\mathbf{r_o}}} \frac{P(\mathbf{r})}{P(\mathbf{o})}\mathbb{I}(\hat{y};a,b,s_1,r_{\hat{Y}})\mathbb{I}(a;w_1,r_A)\mathbb{I}(b;w_0,r_B)\mathbb{I}(w_1;s_1,r_W)\mathbb{I}(w_0;s_0,r_W),$$

and

$$P(\hat{y}_{s_0}|\mathbf{o}) = \sum_{a,b,w,\mathbf{r}\in\mathbf{r_o}} \frac{P(\mathbf{r})}{P(\mathbf{o})}\mathbb{I}(\hat{y};a,b,s_0,r_{\hat{Y}})\mathbb{I}(a;w,r_A)\mathbb{I}(b;w,r_A)\mathbb{I}(w;s_0,r_W).$$

### 4.3 Bounding Path-specific Counterfactual Fairness

In above two sections we express both joint distribution $P(\mathbf{v})$ and the path-specific counterfactual effect as linear functions of $P(\mathbf{r})$. All causal models (represented by different $P(\mathbf{r})$) that agree with the distribution of observational data $\mathcal{D}$ cannot be distinguished and should be considered in bounding PC fairness. Therefore, finding the lower or upper bound of the path-specific counterfactual effect is equivalent to finding the $P(\mathbf{r})$ that minimizes or maximizes the path-specific counterfactual effect, subject to that the derived joint distribution $P(\mathbf{v})$ agrees with the observational distribution $P(\mathcal{D})$. This fact results in the following linear programming problem for deriving the lower/upper bound of path-specific counterfactual effect.

$$\text{min/max} \quad P(\hat{y}_{s_1|\pi,s_0|\bar{\pi}}|\mathbf{o}) - P(\hat{y}_{s_0}|\mathbf{o}), \tag{4}$$
$$\text{s.t.} \quad P(\mathbf{V}) = P(\mathcal{D}), \quad \sum_{\mathbf{r}} P(\mathbf{r}) = 1, \quad P(\mathbf{r}) \geq 0,$$

where $P(\hat{y}_{s_1|\pi,s_0|\bar{\pi}}|\mathbf{o})$ is given by Eq. (2), $P(\hat{y}_{s_0}|\mathbf{o})$ is given by Eq. (3), and $P(\mathbf{v})$ is given by Equation (1).

The lower and upper bounds derived by solving the above optimization problem is guaranteed to be the tightest, since the response function is an equivalent mapping that covers all possible causal models thus we can explicitly traverse all possible causal models.

We use the derived bounds for examining $\tau$-PC fairness: if the upper bound is less than $\tau$ and the lower bound is greater than $-\tau$, then $\tau$-PC fairness must be satisfied; if the upper bound is less than $-\tau$ or the lower bound is greater than $\tau$, $\tau$-PC fairness must not be satisfied; otherwise, it is uncertain and cannot be determined from data.

## 5  Experiments

**Datasets.** For synthetic datasets, we manually build a causal model with complete knowledge of exogenous variables and equations using Tetrad [10] according to the causal graphs. The causal model consists of 4 endogenous variables, $S$, $W$, $A$, $\hat{Y}$, all of which have two domain values. Then, we consider two versions of the causal model: (1) we assume a shared exogenous variables, i.e., a hidden confounder, with 100 domain values (the causal graph is shown in Figure 6); (2) we assume all exogenous variables are mutually independent (the causal graph is omitted due to the space limit). The distribution of exogenous variables and structural equations of endogenous variables are randomly assigned. Finally, we generate two datasets from each version of the causal model, denoted by $\mathcal{D}_1$ and $\mathcal{D}_2$ respectively.

For the real-world dataset, we adopt the Adult dataset, which consists of 65,123 records with 11 attributes including *edu*, *sex*, *income* etc. Similar to [14], we select 7 attributes, binarize their values, and build the causal graph. Fairness threshold $\tau$ is set to 0.1. The datasets and implementation are available at `http://tiny.cc/pc-fairness-code`.

**Bounding Path-specific Counterfactual Fairness.** We use $\mathcal{D}_1$ to validate our method in Eq. (4) for bounding $\text{PCE}_\pi(s^+, s^-|\mathbf{o})$ where $\mathbf{O} = \{S, W, A\}$ and $\pi = \{S \rightarrow W \rightarrow A \rightarrow \hat{Y}, S \rightarrow \hat{Y}\}$. The ground truth can be computed by exactly executing the intervention under given conditions using the complete causal model. The results are shown in Table 2, where the first column indicates the indices of $\mathbf{o}$'s value combinations. As can be seen, the true values of $\text{PCE}_\pi(s^+, s^-|\mathbf{o})$ fall into the range of our bounds for all value combinations of $\mathbf{O}$, which validates our method.

**Comparing with previous bounding methods.** We use $\mathcal{D}_2$ to compare with the previous methods [20, 14] which are derived under the Markovian assumption. We compare with [20] for bounding $\text{PE}_\pi(s^+, s^-)$ with $\pi = \{S \rightarrow W \rightarrow A \rightarrow \hat{Y}, S \rightarrow \hat{Y}\}$. We also compare with [14] for bounding $\text{CE}(s^+, s^-|\mathbf{o})$ with $\mathbf{O} = \{S, W, A\}$. The results are shown in Table 3 where the bold indicates that our method makes different judgments on discrimination detection due to the tighter bounds. As can be seen, our method achieves much tighter bounds than previous methods, which can be used to examine fairness more accurately. For example, when measuring indirect discrimination using $\text{PE}_\pi(s^+, s^-)$ (Row 1 in Table 3), it is uncertain for [20] since the lower and upper bounds are $-0.2605$ and $0.2656$, but our method can guarantee that the decision is discriminatory as the lower

bound $0.1772$ is larger than $\tau = 0.1$. As another example, when measuring counterfactual fairness of the 2nd groups of $\mathbf{o}$ using $\text{CE}(s^+, s^-|\mathbf{o})$ (Row 3 in Table 3), the method in [14] is uncertain since the lower and upper bounds are $-0.4383, -0.0212$ but our method can guarantee that the decision is fair due to the range of $[-0.0783, -0.0212]$.

We also use the Adult datset to compare with the method in [14] for bounding $\text{CE}(s^+, s^-|\mathbf{o})$ with $\mathbf{O} = \{age, edu, marital\text{-}status\}$ and obtain similar results, which are shown in Table 4.

Table 2: Bounds and ground truth of PC fairness on $\mathcal{D}_1$.

| # of o | $\text{PCE}_\pi(s^+, s^-|\mathbf{o})$ | | |
|---|---|---|---|
| | $lb$ | $ub$ | $Truth$ |
| 1 | -0.4548 | 0.5452 | 0.1507 |
| 2 | -0.5565 | 0.4435 | -0.0928 |
| 3 | -0.5065 | 0.4935 | 0.0561 |
| 4 | -0.4598 | 0.5402 | 0.0548 |

Table 3: Compare with existing methods in [20, 14] on $\mathcal{D}_2$.

| | # of o | $Truth$ | Previous methods | | Our method | |
|---|---|---|---|---|---|---|
| | | | $lb$ | $ub$ | $lb$ | $ub$ |
| PE | *N/A* | **0.1793** | **-0.2605** | **0.2656** | **0.1772** | **0.1836** |
| CE | 1 | 0.3438 | 0.0878 | 0.5049 | 0.0878 | 0.5049 |
| | 2 | **-0.0557** | **-0.4383** | **-0.0212** | **-0.0783** | **-0.0212** |
| | 3 | **0.2318** | **-0.1192** | **0.2979** | **0.1282** | **0.2847** |
| | 4 | 0.0800 | -0.2101 | 0.2070 | 0.0110 | 0.1499 |

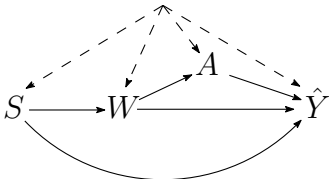

Figure 6: The causal graph for the synthetic dataset $\mathcal{D}_1$.

Table 4: Compare with the existing method in [14] on the Adult dataset.

| # of o | Method in [14] | | Our Method | |
|---|---|---|---|---|
| | $lb$ | $ub$ | $lb$ | $ub$ |
| 0 | 0.0541 | 0.2946 | 0.1498 | 0.1944 |
| 1 | -0.1314 | 0.1091 | -0.1314 | 0.1091 |
| 2 | 0.1878 | 0.3210 | 0.2507 | 0.2890 |
| 3 | -0.0356 | 0.0976 | -0.0356 | 0.0976 |
| 4 | 0.1676 | 0.5289 | 0.4419 | 0.5289 |
| 5 | -0.1634 | 0.1979 | -0.0731 | 0.1979 |
| 6 | 0.1290 | 0.4689 | 0.3942 | 0.4689 |
| 7 | -0.1808 | 0.1591 | 0.0014 | 0.1591 |

## 6    Conclusion

In this paper, we develop a general framework for measuring causality-based fairness. We propose a unified definition that covers most of previous causality-based fairness notions, namely the path-specific counterfactual fairness (PC fairness). Then, we formulate a linear programming problem to bound PC fairness which can produce the tightest possible bounds. Experiments using synthetic and real-world datasets show that, our method can bound causal effects under any unidentifiable situation or combinations, and achieves tighter bounds than previous methods.

As the concern of scalability, the domain size of each response variable is exponential to the number of parents, meaning that the joint domain size of all response variables are exponential to the total in-degree of the causal graph. However, we notice that not all response variables are needed in the formulation, and only those that directly lead to unidentification are needed. For example, when a hidden confounder causes unidentification, only the children of the hidden confounder need to have response variables in the formulation; and when a "kite graph" causes unidentification, only the witness variable need to have a response variable in the formulation. As a result, the total complexity of the problem formulation could be significantly decreased. How to construct fair predictive models based on the derived bounds is another future research direction. One possible method would be to incorporate the bounding formulation into a post-processing method. The new formulation will be a min-max optimization problem, where the optimization variables will include response variables $P(\mathbf{r})$ as well as a post-processing mapping $P(\tilde{y}|\hat{y}, \mathsf{pa}_Y)$. The inner optimization is to maximize the path-specific counterfactual effect to find the upper bound, and the outer optimization is to minimize both the loss function and the upper bound. We will to explore these ideas in the future work.

## Acknowledgments

This work was supported in part by NSF 1646654, 1920920, and 1940093.

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
