[Reviews · NeurIPS 2019]

Reviewer 1



Advantages: 1. The proposed framework subsumes most previous fairness notions. It can potentially help identify, analyze and compare new fairness notions. 2. The proposed bounding method is simple and makes sense. 3. The paper is clear and easy to follow. Weakness: 1. There are many limitations of the proposed method. The proposed method assumes that the causal graphical is given. Also, the values must be discrete. 2. It would be good to show how to use the proposed method to achieve fair policy learning without "severely damaging the performance of predictive model". 3. It would be great to discuss why the fairness bound achieved by the proposed method is tighter compared with previous methods. Minor issues: line 17 irrespective their -> irrespective of their line 240 to find -> to finding Should the the numbers in Table 3 CE #of o 4 be bold? The bound of the proposed method is tighter than previous methods.

Reviewer 2



I was surprised to see that this notion of causal effect has not been defined before. Personally, I always assumed that this notion must be known, but I am unable to find a reference. Therefore, although quite trivial, it seems that this notion of causal effect has not been discussed before in the literature. I may be wrong and advise the authors to do a thorough literature search to verify this. Apart from this, I see this notion as a specific case of counterfactual effect, restricted to a set of paths. The definition should easily follow from Ilya Shpitser's paper which the authors give as well. The linear programming approach for bounding is also known as authors acknowledge. In practice, I am unsure of the impact of this work or this notion. This notion carries all the difficulties of path-based counterfactuals and more, in terms of actually evaluating it, even with interventional data. Line 17: irrespective of etc..->etc. Line 24: total effects of interventions Line 71: we don't make assumption -> we do not make assumptions Please add the relevant references to Section 2. Also, definition 3 is not clear as "performing an intervention along a path" is not formally defined. I recommend the authors to change the abbreviation PC effect as it might mislead the reader into thinking the concept is related to the infamous PC algorithm. Authors should cite "A Potential Outcomes Calculus for Identifying Conditional Path-Specific Effects" by Malinsky et al. The response function representation argued in the lines 164-170 require the assumption that the observed variables have finite support. AFTER REBUTTAL: Thank you for your responses and explaining a real-world use case, please add this to the camera ready. I still strongly recommend changing the title as this title will create a lot of confusion in the causality community. I also want to reiterate that Def. 3 is not mathematically sound, and will be unclear for the readers who are not already familiar with path-specific counterfactual literature, since intervention on a path is not formally defined. Please elaborate on how to actually evaluate this expression mathematically.

Reviewer 3



The authors attempt to unify different definitions of counterfactual fairness frameworks by characterizing them as appropriate conditioning along specific causal pathways. The main unification is simple. However, the primary contribution appears to be a method to bound the fairness effect under the unified definition of path-specific counterfactual fairness by parameterizing the causal model appropriately and mapping the estimation to a constrained optimization. A few things were not clear in this parametrization process. For example the composition in Line 168 only becomes clear later in the examples. In terms of originality, although the method is computationally intense (requiring response variables that scale exponentially as the node degrees), the contribution and originality are useful as they attempt to bound fairness in unidentifiable situations as well. Quality - The paper is technically sound although a lot more description could have been moved and/or included in supplementary material which only includes code and data as of now. This affects clarity of the paper from time to time. The related work seems adequately cited and compared to. Significance - The method is severly limited as it only works for discrete variables and as the authors note, the domain sizes of the response variables can grow significantly and quickly. While they address this to some extent, it is not clear how generalizable this method is to continuous case at all. Nevertheless it is an important contribution to counterfactual fairness literature.

[Author Response · NeurIPS 2019]

We thank the reviewers for the constructive comments. We will revise the paper accordingly. Below are the responses to the main concerns.

**Reviewer #1.**

- The assumption that the causal graph is given is common in the fairness research based on Pearl's structural causal models. In practice, there are quite a number of algorithms to build causal graphs from the data (and possibly some background knowledge), such as the PC algorithm, the GES algorithm, the FCI algorithm, and their variants.

- We admit that it is a limitation of the proposed method that requires all variables are discrete with finite domains. The barrier to continuous variables lies in how to parameterize a causal model for continuous variables with arbitrary distributions and how to solve the infinite-dimensional optimization problem when we estimate the bounds. For the first problem, there are some related work on causal graph learning and inference with continuous variables, under some model/distribution assumptions, e.g. the additive noise model. But relaxing those assumptions is challenging. For the second problem, there will be infinite response variables to parameterize the continuous causal model, thus the optimal solution to $P(\mathbf{r})$ is infinite dimensional. Hence, Eq. 4 (estimate the tight bound) is an infinite-dimensional optimization problem, which is also challenging. How to address these two challenges will be a future direction for our research.

- Constructing fair predictive models is another future research direction. One possible method would be to incorporate the bounding formulation into a post-processing method. The new formulation will be a min-max optimization problem, where the optimization variables will include response variables $P(\mathbf{r})$ as well as a post-processing mapping $P(\tilde{y}|\hat{y}, \mathsf{pa}_Y)$. The inner optimization is to maximize the path-specific counterfactual effect to find the upper bound, and the outer optimization is to minimize both the loss function and the upper bound. We plan to explore this method in future work.

- The proposed method can provide the tightest bounds because the response variables cover all possible domains of $\mathbf{U}$ so that we can explicitly traverse all possible causal models. We will add more explanations about how the proposed method works in the revised version.

- In Table 3, the results of the proposed method are either equivalent to or tighter than previous methods. The bold lines are to highlight the situation where the tighter bounds make differences in detecting discrimination, showing the practical meaning of the proposed method.

**Reviewer #2.**

- To the best of our knowledge, the notion of path-specific counterfactual effect has not been proposed in previou works. It is worthy to point out that a similar term has been used in paper "Path-Specific Counterfactual Fairness" (AAAI'19), but with a different meaning. The paper studied the causal effect along some specific pathways without conditioning on any observed values, which is equivalent to path-specific fairness, a special case of our proposed fairness notion where $\mathbf{O} = \emptyset$. In paper "A Potential Outcomes Calculus for Identifying Conditional Path-Specific Effects" (AISTATS'19), the conditional path-specific effect is different from our notion in that, for the former the condition is on the post-intervention distribution, and for the latter, the condition is on the pre-intervention distribution. We will add more references and discussions in the revised version.

- Our proposed notion is definitely practical. It can not only unify the previous notions but also resolve new types of fairness that the previous notions cannot do. A typical example is individual indirect discrimination, which means discrimination along the indirect paths for a particular individual. Individual indirect discrimination has not been studied yet in the literature, probably due to the difficulty in definition and identification. However, it can be directly defined and analyzed using our proposed notion by letting $\mathbf{O} = \{S, \mathbf{X}\}$ and $\pi = \pi_i$. Note that the condition here is on the pre-intervention distribution, i.e., we focus on a particular individual with certain observed values, and want to estimate the change of these values after the intervention is performed. Thus, individual indirect discrimination cannot be defined using the above conditional path-specific effect. We will add the above discussions and make our motivation clearer in the revised version.

**Reviewer #3.**

Thanks for the comments. We will incorporate all the comments into the revised version. We will add more deriving details for Section 4.2, reorganize this manuscript accordingly, and move some discussions into the supplementary file if necessary.

[Meta-Review · NeurIPS 2019]

While reviewers agree that the proposed framework is an important addition in unifying notions of counterfactual fairness, reviewers point out limitations that could reduce its practical impact. While reviewers agree that the authors address this to some extent, more discussion of the limitations would be good to add for the camera ready version (along with various revisions suggested by the reviewers including re-organizing the paper to make it clearer). I agree with R2 in recommending that the authors change the abbreviation of PC effect.